# A Follow up on the Continuum Theory of Eco-Anxiety: Analysis of the Climate Change Anxiety Scale Using Item Response Theory among French Speaking Population

**DOI:** 10.3390/ijerph21091158

**Published:** 2024-08-30

**Authors:** Taha Hannachi, Sonya Yakimova, Alain Somat

**Affiliations:** Laboratory of Psychology, Cognition, Behaviour, Communication (LP3C), Department of Psychology, Faculty of Human Science, Rennes 2 University, 35000 Rennes, France; sonya.yakimova@univ-rennes2.fr (S.Y.); alain.somat@univ-rennes2.fr (A.S.)

**Keywords:** eco-anxiety, item response theory, continuum, climate change anxiety scale

## Abstract

The mental health impact of the environmental crisis, particularly eco-anxiety, is a growing research topic whose measurement still lacks consensus. This study aims to use item response theory (IRT) to gain a deeper understanding of the constructs measured by existing questionnaires. To conduct this review, we applied the graded response model with the help of the MIRT package in R on open-access data from the short French version of the Climate Change Anxiety Questionnaire, which measures cognitive-emotional impairment and functional impairment. The models tested in this study are the one, two, and three-factor models, and the bifactor model. After model selection, the psychometric properties of the selected model were tested. Our results suggest that the unidimensional model seems to be the most appropriate for measuring eco-anxiety. The item difficulty parameter extracted from the IRT enabled us to discuss the severity levels of the items comprising this tool. The Climate Change Anxiety Questionnaire appears to be more appropriate for measuring moderate to severe eco-anxiety. Avenues for improving this questionnaire and the measurement of eco-anxiety in general are then discussed.

## 1. Introduction

### 1.1. The Continuum Hypothesis of Eco-Anxiety

For several years now, there has been growing interest in the impact of the ecological crisis on mental health. The term often used to describe this phenomenon is eco-anxiety. However, its definition may vary depending on the context of the study and the concepts used to quantify it [1,2,3]. What is common, however, is the idea that eco-anxiety is a form of psychological distress of varying significance [3,4,5,6], ranging from severe distress to what might be described as worry or concern. Thus, eco-anxiety can be situated along a continuum ranging from mild forms of eco-anxiety to more severe forms, generating emotional, cognitive, social, and/or functional difficulties [3,4,7,8]. The term eco-anxiety is often used interchangeably with climate change anxiety, but some researchers choose to distinguish the two [1,3,5]. For instance, Hogg et al. [5] consider that eco-anxiety encompasses the psychological distress related to the ecological crisis in its entirety, including climate change anxiety, which covers anxiety related to anthropogenic climate change. It also encompasses distress from “a multiplicity of environmental calamities, which may or may not be directly caused by climate change, including the elimination of entire ecosystems and plant and animal species, global mass pollution and deforestation” ([5], p. 1). While theoretically the two concepts might be different, there is not enough empirical evidence to suggest that they should be distinguished. In this article, we use the terms “eco-anxiety” and “climate change anxiety” (or “climate anxiety” for short) interchangeably because we believe that, in all cases, eco-anxiety fundamentally stems from the perception of the human activity’s negative impact, regardless of the specific consequences.

Many authors seem to focus on the most severe forms of eco-anxiety, adopting a pathologizing approach to the concept [9] rather than adopting a continuous perspective that allows climate anxiety to be viewed as a continuum of reactions ranging from mild to severe reactions to the ecological crisis [2,10]. Indeed, even the mildest forms of climate anxiety can have motivational consequences, prompting people to adopt a more eco-responsible lifestyle [11].

Attempting to quantify eco-anxiety is relatively a new task [4,5]. Thus, not all measurement tools incorporate in their development the idea of a continuous conception of eco-anxiety. For example, Lutz et al. [2] demonstrate that the Climate Change Anxiety Scale (CCAS, [4]), one of the most widely used tools in the world, only assesses severe manifestations of eco-anxiety, thus serving as a clinical (even psychologizing) approach to identifying people in need of care.

Several authors have attempted to quantify this psychological discomfort linked to climate change and the environmental crisis in other ways, but no conceptualization takes into account the full spectrum of this discomfort [4,5,12,13]. Beyond the clinical aspect assessed by the CCAS, the psychological reaction to the environmental crisis is assessed either as worry via the Climate Change Worry Scale [12] or as distress and impairment by the Climate Change Distress and Impairment Scale [13]. The Hogg Eco-Anxiety Scale (HEAS, [5]) claims to measure non-pathological eco-anxiety and represents an alternative to the CCAS. However, this questionnaire adapts items from the GAD-7, a tool designed to measure generalized anxiety disorder and considered by Jordan et al. [14] to be primarily suited for diagnosing severe anxiety. Thus, HEAS scores appear to be highly correlated with CCAS scores [15]. Furthermore, these two questionnaires have the same floor effect problem. For both the CCAS [4,16] and the HEAS [5,15], the means reported (with comparable standard deviations slightly below 1) are close to the lower limit of the Likert scale used, with the exception of the last factor of the HEAS. This makes these scales particularly poor at detecting mild forms of eco-anxiety. In summary, while these questionnaires—particularly the CCAS—may be effective in identifying individuals with severe forms of eco-anxiety who require psychotherapeutic intervention due to the physiological, cognitive, and functional symptoms they describe, they are less effective in detecting mild cases that may benefit from other types of interventions. Given the widespread use of the CCAS, applying it to the general population could be counterproductive. This study aims to explore this issue further and offers insights into potential improvements for such a scale.

Logically, if we distinguish between two extremes (weak and strong), intermediate stages of eco-anxiety should also exist. The only attempt to encompass both “ends” of the concept within a single questionnaire is made by Àgoston et al. [17]. The authors pointed out two factors representing the extremes: habitual ecological worry and the negative consequences of eco-anxiety. However, the factor analysis fails to identify a factor that captures an intermediate stage of eco-anxiety.

Consequently, we believe it is important to develop a tool that encompasses the entire spectrum of anxiety caused by climate change. Such a tool would enable a better understanding of the reasons behind the variability in eco-anxiety between individuals (inter variability), or even for the same individual, depending on the context and the time of measurement (intra variability).

This article represents a first step in exploring the spectrum of eco-anxiety. By applying statistical methods derived from item response theory (IRT) [18], we aim to examine the factor structure of the CCAS [4]. We hypothesize that the authors’ original approach (factor analysis) may hinder the expression of a symptom severity hierarchy. To test this hypothesis, we will adopt a different statistical perspective (IRT), which allows for conceptualizing eco-anxiety as a continuum. This approach likely aligns more closely with the reality of the distress people experience regarding the ecological crisis, as it enables us to describe the progression of the phenomenon while preserving the scaling properties of the questionnaire.

### 1.2. Item Response Theory

It is very conventional to consider the ordinal categorical modalities of a Likert scale as numerical and continuous, in order to be able to apply linear modeling (principal component analysis). This assumption of linearity can, in the case of principal component analysis, give rise to completely artifactual factors (for a detailed critique on principal component analysis, see [19,20]). In contrast to the frequently used linear model, item response theory (IRT) takes into account the fact that response modalities are categorical and bounded. They are categorial in a sense that the scale is discrete, not continuous, and bounded in the sense that it is finite, not infinite, like the linear model assumes the data to be. Moreover, unlike conventional testing methods, IRT distinguishes between an item’s level of difficulty and its ability to discriminate between individuals. More precisely, IRT stipulates that the probability of an individual responding in an extreme manner to an item (the indicator) depends on his or her ability to endorse the phenomenon (latent trait) described by that item. This approach allows IRT to standardize the level of difficulty of each item on the same scale based on people’s ability to endorse them, enabling direct comparisons. Consequently, two individuals with comparable scores according to factor analyses may not actually be at the same level of endorsement of the studied phenomenon, depending on the difficulty level of the items.

De Boeck [21] demonstrates, for example, in relation to the use of psychological tests, that the scores of a person’s ability to endorse the latent trait produced by IRT models offer us the possibility of identifying a hierarchy of symptoms experienced by people. Factor analyses, on the other hand, are based on averages, which erase information on how participants responded to the items that are most diagnostic of the phenomenon, thus preventing us from identifying the most symptomatic participants.

In the specific case of eco-anxiety, for example, it is claimed that eight out of ten French people suffer from eco-anxiety [22]. IRT provides a solution to this examination by projecting individuals’ latent trait scores onto the same graphical space as the difficulty level of the proposed items. This method thus provides a description of the phenomenon that is closer to reality (see [23]).

Furthermore, according to Clayton and Karazsia [4], there is no clear threshold for the transition from adaptive climate anxiety to maladaptive eco-anxiety. They argue that a person with an overall CCAS score above the median is one for whom climate change is having a significant impact on their mental health. Hogg et al. [5] challenge this proposal, believing that only people with extreme overall scores on the HEAS should be considered as suffering from severe eco-anxiety.

### 1.3. The Factor Structure of the CCAS: A Lack of Consensus

In its initial validation, the CCAS is a 22-item scale that describes four dimensions: cognitive and emotional difficulties, functional difficulties, experience (direct or indirect of climate change), and environmental engagement. Researchers refer to the first two as measuring eco-anxiety and the last two as control factors [4]. However, several studies have shown that removing the last two dimensions improves the psychometric quality of the scale (e.g., [16,24]). We therefore follow these recommendations and use only the first 13 items of the scale, which cover the dimensions of cognitive and emotional impairment, and functional impairment, which we will refer to as CCAS-13.

In their article validating the CCAS in French, Mouguiama-Daouda et al. [16] carried out only confirmatory factor analyses to test the one- or two-factor structure of the CCAS-13 and the four-dimensional structure (22-item version). They retained the two-factor structure of the 13-item version. We aim to confirm this structure of the CCAS-13 by checking whether there are other models that fit the data better. According to Immekus et al. [25], in addition to theoretical and empirical data, the choice of instrument model must be made following exclusion of the presence of other models that could lead to an alternative explanation of the factor structure. Cruz and High [26] explored a second-order model for the CCAS-13. They found that both CCAS-13 factors are summarized by an overall second-order factor, i.e., they measure the same construct. Thus, they recommend the use of a hierarchical structure (one global factor and two first-order factors). On the other hand, Larionow et al. [27] found that a three-factor solution was better than the two-factor solution in the Polish context, although they did not report the significance of this difference. They also observed that correlations between factors are relatively high and recommend the use of global scale scores. In a sample of participants from three Asian countries and the USA, Tam et al. [28] demonstrate that a single-factor model does not have a good fit and argue that the alternative two-factor solution is probably better. Nevertheless, the inter-factor correlations for the latter solution remain very high (this is true for each of the four countries represented in this study) and suggest the existence of a hierarchical structure, such as that mentioned by Cruz and High [26] and Larionow et al. [27]. To summarize, a lot of models tested on the CCAS and CCAS-13 seem to be plausible but not consensual.

The aim of this article is to provide some keys to clarifying these fuzzy areas relating to the conceptualization and measurement of eco-anxiety through the use of IRT. Thus, we want to explore alternative models that may have a better fit for the data and may provide a more plausible explanation of the eco-anxiety continuum.

## 2. Materials and Methods

This study utilized pre-existing datasets that were publicly available online. A total of 1778 responses to the French version of the CCAS were retrieved. The first dataset originated from study two by Mouguiama-Daouda et al. [16], and the second dataset originiated from the study by Heeren et al. [29]. The data for the 905 respondents in Mouguiama-Daouda et al.’s [16] study two were obtained from https://osf.io/m3ygz/ (accessed on 15 November 2022), while the data for 873 respondents in Heeren et al. [29] were retrieved from https://osf.io/2r659/ (accessed on 15 November 2022). These datasets were made available by the researchers under the “CC-BY Attribution 4.0 International” license. The two datasets were selected for this study because they are comparable in size, were collected in the same context, and employed similar methods of data collection. In the original studies, data were gathered via social media and listserv advertisements, targeting a French-speaking population. The participants were predominantly European, with a small minority (less than 5%) from other French-speaking regions.

The short version of the CCAS (13 items) is evaluated in this article; hence, the acronym CCAS-13 is used. According to its developers, the CCAS-13 comprises eight items measuring “cognitive and emotional impairment” and five items measuring “functional impairment”. Responses are based on a five-point Likert-type frequency scale (1 = “never”, 2 = “rarely”, 3 = “sometimes”, 4 = “often” and 5 = “always”).

The data analysis was performed with the MIRT (multidimensional item response theory) package [30] on R software [31], version 4.3.0. The graded response model (GRM) [32] was used. This model is suitable for the analysis of ordinal data, particularly Likert scales [33]. It is also more flexible than other IRT models in that it allows for the discrimination parameter to be estimated for each item [34]. The estimation method used is the Expectation-Maximization (EM) algorithm, which is used to maximize the likelihood of the model parameters given the observed data [35]. The Oakes method was used as a type of estimation method to calculate the parameter information matrix for computing the standard error (see Chalmers, [36] for details). The item type used was of course “graded” for the GRM.

First, we revisit the unidimensional model, the two-factor model, and the three-factor model to compare the fit indicators given by IRT with the results of previous research. To explain the strong correlations between the CCAS-13 dimensions found in the literature, we will also test the bifactor model [37]. According to Gibbons and Cai [38] (p. 52), the latter is more appropriate for “self-report measures of health status covering both physical and emotional difficulties”. This model compiles a general factor of the phenomenon and the uncorrelated sub-domains that make it up. It is therefore more flexible than the hierarchical model in explaining correlated factor structures [39]. Indeed, the bifactor model allows us to decide on the relevance of sub-domain scores by looking at the loading force not absorbed by the general factor [40].

Figure 1 shows the different models tested in this article.

Statistical analyses of the data were carried out in two stages: (1) tests of goodness of fit of models and (2) verification of the psychometric properties of the chosen model.

For the first stage, four models were tested: the unidimensional model, the multidimensional two- and three-factor model, and the bifactorial model. The model fit was tested using the limited information goodness-of-fit statistic *M*_2_ [41]. This test is more appropriate than the Pearson’s test statistic χ^2^, especially in the context of IRT models [41]. The *M*_2_ test belongs to the limited information test category. These tests were designed to solve the problem faced by full information tests like the χ^2^; they depend on all the information in the contingency table, especially in models with many variables, where the probability of combining certain categories is becoming increasingly rare. Indeed, according to Steinberg and Thissen [42], for a number of categories greater than or equal to five, the fit approximation becomes invalid for any model as soon as the number of items exceeds six, regardless of sample size. One of the major interests of IRT is to tackle the problem of probability sparsity by examining the sample in terms of how participants responded to difficult answers [43]. The idea behind limited-information tests is to reduce the dependence on probabilities for rare response combinations [44], which can improve the power of the tests and the accuracy of the model fit assessment. Probability boxes removed from the contingency table are, in fact, more likely to appear (see [45] for more on the difference between limited-information and full-information tests).

Apart from the *M*_2_ test, we also consulted the values of other important indicators. In particular, we followed the recommendations of Maydeu-Olivares and Joe [46], according to which an RMSEA (Root Mean Square Error of Approximation) of less than 0.089 and a SRMR (Standardized Root Mean Square Residual) of less than 0.05 are acceptable model fits for the IRT. We also took into account the Comparative Fit Index (CFI) and the Tucker Lewis Index (TLI). They range from 0 to 1, indicating that model fit is acceptable when their respective values are above 0.90, and that model fit is excellent when their respective values are above 0.95 [47,48]. We have also reported the values of the Akaike Information Criterion (AIC) [49]. This indicator, which incorporates complexity and level of fit into its equation, is useful when selecting the best model from among several plausible ones [50]. The lower the value of the AIC compared to previous models, the better the model is [51].

Once the most suitable model had been selected, the psychometric qualities of the chosen model were checked. For each item, we evaluated the parameters of discrimination (noted *a*) and difficulty (noted *b*_1_, *b*_2_, *b*_3_, and *b*_4_, respectively, relative to the 50% probability of choosing “rarely” over “never”, then “sometimes” over “rarely”, then “often” over “sometimes”, and finally “always” over “often”). Based on the principle that, according to IRT, the latent trait is represented by a continuum going from a low level to a high level, the discrimination parameter (*a*) determines how well an item is able to discriminate individuals along the latent trait continuum [52]. According to Baker [53], item discrimination becomes acceptable when the *a* value is greater than or equal to 0.5. Difficulty parameters (*b*_1...4_) are based on the thresholds at which an individual has a 50% probability or more of selecting any category of an item [54]. The *b*_1...4_ values are standardized on a scale ranging, in our case, from −6 to 6. This choice was made in order to be able to graphically represent the most difficult modalities of the scale with values greater than 3. A *b* value close to −6 reflects the ease of endorsement of the item category in question. Conversely, an item category with a *b* value close to 6 is difficult to endorse.

One of the advantages of IRT is the independence of the estimated item parameters from the sample, and therefore the relative consistency of estimates across samples. Thus, a well-fitted model will always return the same item parameter estimates, regardless of the sample [23]. As a result, the same analyses were carried out separately on the two datasets, with no major differences. Consequently, we will only present the results obtained on a single dataset from the work of Mouguiama-Daouda et al. [16] (*n* = 905). The full analyses carried out on both datasets are presented in the Appendix A.

## 3. Results

### 3.1. Model Selection

The results show that model fit statistics are acceptable for all the models tested, with a slightly higher value of the threshold for SRMR for the one-factor model and the two-factor model. All detailed results are displayed in Table 1. The model comparison (Table 2) shows that the two-factor model is significantly better than the one-factor model: χ^2^ (1) = 191.234, *p* < 0.001. similarly, the three-factor model is significantly better than the two-factor model: χ^2^ (2) = 232.094, *p* < 0.001. However, we find no significant difference between the three-factor model and the bifactor model. Though, the bifactor model is, like the three-factor model, significantly better than the one-factor and two-factor model. This result is confirmed by the consecutive decrease in the value of the AIC (the decrease in the AIC value is not as important when comparing the three-factor model to the bifactor model as in the previous cases).

For the one-factor model and the two-factor model, the results of the IRT-based dimensionality analysis are comparable to the results of the confirmatory factor analysis of study two by Mouguiama-Daouda et al. [16]. Moreover, the correlation between the two factors (cognitive and emotional impairment (CEI) and functional impairment (FI)) is even stronger at 0.83, compared with 0.66 in Mouguiama-Daouda et al. [16].

This result consolidates Cruz and High’s [26] assertion that these two factors are not independent of each other; they could therefore be measuring the same construct: eco-anxiety. This leads us to question the relevance of this two-factor model, which could be modeled differently.

We therefore tested a three-factor model by carrying out an exploratory factor analysis, the results of which are also presented in Table 1.

The *M*_2_ goodness-of-fit test of the three-dimensional model is insignificant. We therefore retain the null hypothesis that this three-factor model fits the data perfectly. The other indicators (RMSEA, SRMSR, etc.) confirm this finding. Moreover, this model is significantly better than the other two (Table 2).

In fact, the cognitive and emotional difficulties (CEI) factor is divided into two (potentially, physiological symptoms (SYM) and rumination (RUM)).

This three-factor modelling (Table 3) is more faithful to the questionnaire construction process (the first 13 items) adopted by Clayton and Karazsia [4]. As a reminder, the first four items, which correspond here to the SYM factor, were constructed on the basis of reading the scientific literature, but also of people’s experiences shared on blogs. The aim of these four items is to reflect “physical symptoms”. The next four items belonging to the RUM dimension were indeed adapted from the rumination questionnaire by Treynor et al. [55]. As for the last five (from the CCAS-13 version), representing the functional impairment factor, they were adapted from the Weiss Functional Impairment Rating Scale [56]. This is in line with the three-factor structure validated by Larionow et al. [27] in their Polish-language validation of the CCAS-13.

Despite its perfect fit to the data, this model presents two questionable points. The first concerns the correlations between factors, which remain relatively strong (Table 3: factors correlation). This suggests that the different factors measure almost the same thing. Second, the variance explained by the three-factor model with rotation (55.6%) is not much higher than the variance explained by the unidimensional model (46%). Without rotation, the first factor of the three-factor model explains 41.4% of the variance (Table 4), while the second and third explain only 7.4% and 7.3% of the total variance, respectively. The difference of explained variance between the rotated and the unrotated models might partly be due to the choice of the oblique rotation. The problematic use of rotation in factor analysis and interpretation will be discussed below.

The bifactorial model, the last model we tested, is likely to provide more of an explanation of the strong correlations between the previously extracted factors. The results show that it fits the data significantly better than the unidimensional model, in contrast to what Cruz and High [26] found. However, it does not show a significant improvement over the three-dimensional model (see Table 2).

However, it is important to note that the general factor absorbs all item loading. The loadings of the scale’s sub-domains are low, and even negative for some items. Furthermore, most of the explained variance of the two-factor model is attributed to the global factor (Table 5). This translates into an explained variance of the general factor of the bifactor model comparable to that of the unidimensional model (Table 6). We therefore confirm the recommendation put forward by Cruz and High [26], inviting researchers and practitioners to mainly report the global scores of the CCAS. In line with what these authors suggest, our analyses show that the CEI and FI factor scores are somewhat irrelevant.

Despite the fact that all the models that we tested have acceptable fit statistics, they all have some issues, except for the one-factor model. As seen before, the two and three-factor models suffer from a high correlation value between their factors. Additionally, the bifactor model has weak loadings on its sub-domains, which make their computation and interpretation irrelevant. Thus, it appears that the unidimensional model is more appropriate for measuring climate anxiety by the CCAS. From a theoretical point of view, this choice seems more consistent with the idea of a continuum of eco-anxiety. That is why we concentrate the exploration of the one-factor model’s psychometric properties.

### 3.2. Psychometric Properties of the CCAS-13 Unidimensional Model

Overall, all items have a satisfactory discrimination index (*a*) (Table 7). However, item 7 has an *a*-value of less than 1, which means that its wording needs to be readjusted or reformulated. Although acceptable, the discrimination value of this item is not as high as that of the other items, especially as item 7 has proved problematic in several other studies [16,26,27].

The IRT classifies the response modalities of each item on the same axis by calculating threshold values corresponding to a 50% probability of choosing a given modality (Table 6 and Figure 2). It also calculates a general difficulty value for each item (Figure 3).

The overall difficulty level of the items compared with participants’ scores on the latent trait (Figure 3) shows that the CCAS-13 mainly provides information on high levels of eco-anxiety. As a result, the questionnaire provides little information on almost half the sample, who identify with little or none of the current CCAS-13 items. For these participants, the CCAS-13 points to an absence of symptoms as described by the items, but tells us nothing about their mental state in relation to climate change.

## 4. Discussion

This study offers novel insights into the measurement of climate anxiety by challenging the existing factor structures of the CCAS-13. By critically analyzing the limitations of multi-factor models and proposing a unidimensional approach, we provide a fresh perspective on how eco-anxiety should be conceptualized and measured. For that purpose, we explored alternative CCAS-13 factor structures that may better explain climate anxiety. We argued in favor of using the one-dimensional model rather than the two- or three-factor model of the CCAS-13. Our results are in line with what has been found in the literature about the goodness of fit of the unidimensional model [16,26]. In fact, values of RMSEA in the literature range from 0.17 to 0.08 for the unidimensional model of CCAS-13 (see Hogg et al. [24] for a synthesis of tested models in the literature) against a value of 0.08 in this study. The SRMR values range from 0.07 to 0.04 against 0.074 for this study.

Indeed, the two- and three-dimensional models in this study, but especially in previous ones, present a problem of strong correlations between their dimensions. This means that the different factors measure almost the same thing. This misinterpretation might partly be due to the almost systematic choice of oblique rotations [57].

Although rotation simplifies model selection and interpretation, it can also be misleading [57]. Factors interpreted a posteriori are merely the result of the chosen rotation. This choice of rotation exacerbates what Van Schuur and Kiers [20] call the “extra factor” phenomenon in exploratory factor analyses. These authors describe how, for a potentially bipolar concept, factor analyses misinterpret the (potentially quadratic) relationship between the latent dimension and its indicators, transforming what should be a unidimensional bipolar phenomenon into a multidimensional one, the factors of which are more or less independent. Furthermore, the discussion around the eco-anxiety continuum, and the distinction between practical eco-anxiety and paralyzing eco-anxiety, raised questions about the multi-factor modeling of this construct. A recurring feature of the CCAS-13 data modelling we carried out without axis rotation is that there is always a dominant factor that explains most of the variance on its own. The bifactor model, which we also tested, and which proposes a solution to circumvent the use of rotations, confirms that a general climate anxiety factor is the most relevant in the case of CCAS-13. Our critique of the common use of oblique rotations highlights a significant methodological concern that has not been adequately addressed in previous studies. This critique is crucial for future research, as it raises awareness about potential misinterpretations because of this practice, leading to more accurate and reliable measurement tools. Additionally, the introduction of the bifactor model not only confirms the relevance of a general climate anxiety factor but also offers a practical solution to the limitations of traditional factor analysis techniques. This approach can be instrumental for researchers and practitioners in developing more streamlined and effective assessment tools.

We have also demonstrated the usefulness of the IRT method in providing new perspectives on the conceptualization of eco-anxiety. It consolidates the tool, but also opens up prospects for its improvement. Indeed, the analysis results of the unidimensional model using IRT show that the positions of the items in terms of difficulty are similar across data sets, thus indicating a good consistency of the eco-anxiety construct as described by CCAS-13. Consequently, in future studies, it would be more interesting to validate tools based on a hierarchy of symptom severity.

In this regard, we note that items from the two dimensions (CEI and FI) defined by Clayton and Karazsia [4] overlap on the eco-anxiety continuum. By interpreting the position of the items and analyzing their content in depth, we can begin to trace a continuum of the place eco-anxiety takes in a person’s life. According to the results, it starts with (1) having problems managing one’s needs and interest in the sustainable; (2) questioning the way the environmental situation is handled cognitively and emotionally; (3) having social problems and a loss of meaning at work; and (4) falling into despair, accompanied by severe physical symptoms such as crying and nightmares.

In itself, the CCAS-13 serves the authors’ objective [4] of identifying severe to clinical cases of climate change anxiety. However, one part of the continuum remains unexplored: the part which concerns people with low scores. This limits the use of the CCAS among the general public and raises questions about how this issue is represented.

Based on the floor effects observed on the distributions of the various items, states of eco-anxiety other than those described above are likely to be grouped in the non-anxious category. However, even if these people do not suffer from severe symptoms, they may nonetheless feel a certain amount of anxiety and stress relative to the perceived environmental crisis. In a recent qualitative study, Marczak et al. [11] show that people who are strongly affected by the ecological crisis may not present any of the symptoms described in the CCAS-13 [11]. If this questionnaire is to be adapted to a general population, we need to consider adjusting and supplementing it by including items that refer to less severe manifestations, and thus formulate a tool that could fit in with the idea of a continuum relating to eco-anxiety.

A number of researchers are now advocating the continuum hypothesis of climate anxiety [2,7], which makes it possible to identify people who are worried, stressed, or even anxious about climate change, even though they do not display any manifestations measured by the CCAS or HEAS. In the current state of the literature, we have several tools that can be used to identify different levels of discomfort with climate change (i.e., Climate Change Worry Scale, Climate Change Distress Scale, HEAS). Combining these measurement instruments with the CCAS would support the idea suggested by Mathé et al. [15] of developing a unified measurement scale covering the entire eco-anxiety continuum. The availability of such a tool would make it possible to not only care for people suffering from eco-anxiety, which they are unable to manage on their own, but also to offer others the opportunity to engage in action to enable them to manage the discomfort they feel on their own. The creation of such a scale may help mental health professionals adapt their intervention to the level of eco-anxiety of the patients, and also help behavioral change experts and policy makers to take into account the different levels of eco-anxiety of individuals in studying the social acceptability of new climate policies. This study has a broader objective, which is the advocation for a non-pathologizing approach to eco-anxiety, as, in our point of view, eco-anxiety is a natural outcome for the lack of sense of collective agency and the ability to initiate systemic change while perceiving the need for urgent societal change.

### 4.1. Limitations

This study is not without limitations. First, participants questioned in the samples are mainly French-speaking Europeans. However, a comparison is made with modeling conducted on an American population. In no way can we guarantee the absence of a cultural effect on modeling. Second, the analysis package for bifactorial models is limited. As a result, certain tests could not be carried out. So, the evaluation of the bifactorial model should be carried out again when the appropriate tools are available, since according to Gibbons and Cai [38], the bifactorial model is more promising due to its greater accuracy, especially in the context of supposedly multidimensional data.

### 4.2. Perspectives

In order to validate a questionnaire that could be used with the entire population, we would have to resort to “computer adaptive testing” [58]. This method would require the development of a large panel of questions based on the CCAS-13 items, but also consider what a more moderate level of eco-anxiety encompasses. In this way, we could better identify the threshold for distinguishing between the pathological and motivational dimensions of eco-anxiety relating to the ecological transition [11]. The use of other statistical methodologies, such as the Receiver Operating Characteristic (ROC) [59], could enable the definition of thresholds and optimally detect false positives.

## 5. Conclusions

This study aimed to explore the structure of the Climate Change Anxiety Scale (CCAS-13) using item response theory (IRT), offering new insights into the measurement of eco-anxiety. Our analysis demonstrated that the unidimensional model, which assumes a single underlying factor, provided the best solution for modeling eco-anxiety. These findings challenge previous multi-dimensional models and support the notion that eco-anxiety operates along a single continuum from mild concern to severe anxiety. The IRT analysis also revealed that the CCAS-13 is more effective at detecting severe forms of eco-anxiety, with less sensitivity to milder manifestations.

These results have significant implications for both research and clinical practice. For researchers, our findings suggest that the current tools might be insufficient for capturing the full spectrum of eco-anxiety, particularly its milder forms. This highlights the need for the development of new measurement tools or the refinement of existing ones to ensure they are comprehensive and sensitive to varying levels of eco-anxiety. Clinically, the results underscore the importance of considering the continuum of eco-anxiety when assessing individuals, as the current focus on severe symptoms may overlook those experiencing less intense, yet still impactful, eco-anxiety.

Future research should explore the development of more nuanced instruments that can capture the entire eco-anxiety spectrum and further investigate the factors that contribute to the variability in eco-anxiety experiences across different populations. Understanding these dynamics will be crucial for both improving mental health interventions and fostering more effective public communication strategies about ecological transition policies.

## Figures and Tables

**Figure 1 ijerph-21-01158-f001:**
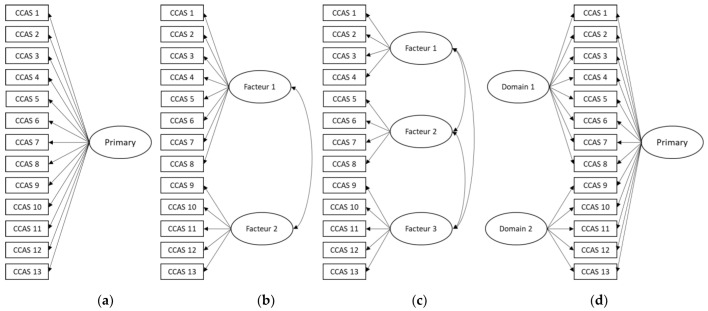
The four models tested in this study: (**a**) unidimensional model, (**b**) correlated 2 factor model, (**c**) correlated 3 factor model, and (**d**) bifactor model.

**Figure 2 ijerph-21-01158-f002:**
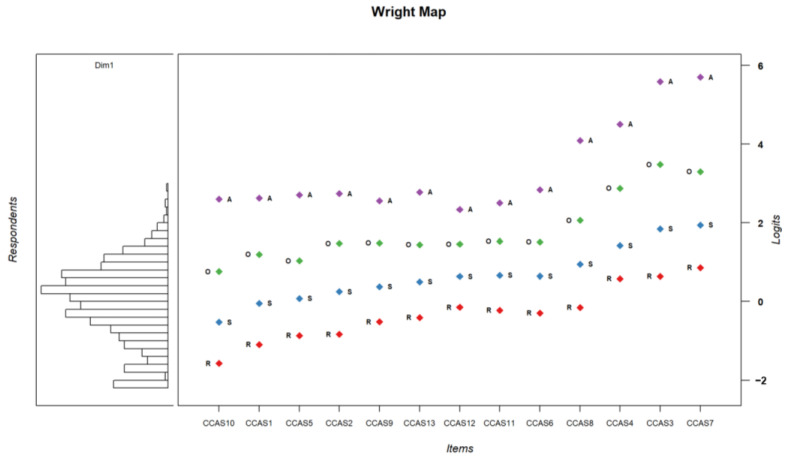
Projection of thresholds for the modalities of each item and participants’ scores standardized on the same scale. **Right panel** represents participants’ scores on the eco-anxiety dimension. **Left panel** represents thresholds for having a 50% chance of endorsing a particular modality with A = always, O = often, S = sometimes, R = rarely.

**Figure 3 ijerph-21-01158-f003:**
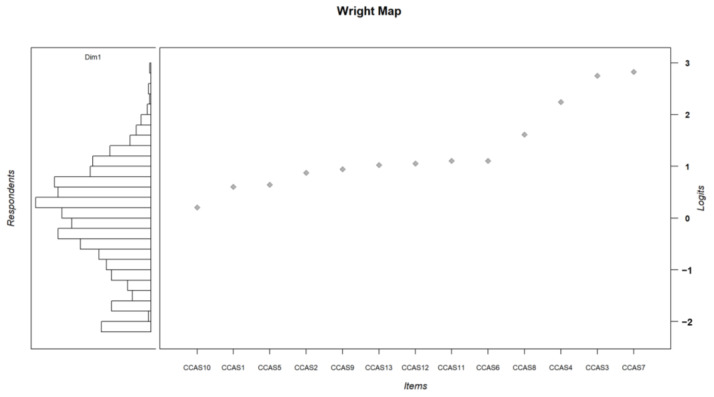
Wright map: comparison between persons’ scores and general items difficulty. **Right panel**: items are sorted by their general level of difficulty. **Left panel**: persons’ latent trait level of eco-anxiety.

**Table 1 ijerph-21-01158-t001:** Comparison of the fit of CCAS-13 models between the work of Mouguiama-Daouda et al. [16] and the present work. Model Fit Tests Performed by Mouguiama-Daouda et al. [16].

Model	χ^2^	df	*p*	RMSEA	SRMR	TLI	CFI	AIC
1 factor	542.26	65	<0.01	0.10	0.06	0.81	0.84	31,355.91
2 factors	390.48	64	<0.01	0.08	0.05	0.87	0.89	31,170.33
Model fit tests carried out in this work.								
**Model**	** *M* _2_ **	**df**	** *p* **	**RMSEA**	**SRMSR**	**TLI**	**CFI**	**AIC**
1 factor	176.38	26	<0.001	0.08	0.074	0.91	0.94	27,402.92
2 factors	173.14	25	<0.001	0.08	0.07	0.90	0.94	27,213.69
3 factors	3.46	3	0.33	0.01	0.04	0.99	0.99	26,974.11
bifactor	13.89	13	0.38	0.008	0.049	0.99	0.99	26,959.97

χ2: Pearson’s test statistic, *M*_2_: the limited information goodness-of-fit test statistic, df: degree of freedom, *p*: *p* value, RMSEA: Root Mean Square Error Approximation, SRMSR: Standardized Root Mean Squared Residual, TLI: Tucker-Lewis Index, CFI: Comparative Fit Index, AIC: Akaike Information Criterion.

**Table 2 ijerph-21-01158-t002:** Comparison between the 1 to 2 and 3 dimensional models and the two-factor model.

Model	AIC	χ^2^	df	*p*
1 factor	27,402.92			
2 factors	27,213.69	191.234	1	<0.001
3 factors (Exploratory)	26,985.59	232.094	2	<0.001
Bifactor	26,959.97	5.857	10	0.82

**Table 3 ijerph-21-01158-t003:** Three-factor model item loading (with Oblimin rotation).

	FI	SYM	RUM
CCAS1	-	0.58	-
CCAS2	-	0.94	-
CCAS3	-	0.69	-
CCAS4	-	0.56	-
CCAS5	-	-	0.65
CCAS6	-	-	0.75
CCAS7	-	-	0.43
CCAS8	-	-	0.76
CCAS9	0.80	-	-
CCAS10	0.57	-	-
CCAS11	0.86	-	-
CCAS12	0.81	-	-
CCAS13	0.61	-	-
Factor correlations			
	**FI**	**SYM**	**RUM**
FI	-		
SYM	0.650	-	
RUM	0.736	0.609	-
Explained variance			
	**FI**	**SYM**	**RUM**
Explained variance	22.2%	18.3%	15.1%

**Table 4 ijerph-21-01158-t004:** Comparison of factor loadings for the 3-dimensional model without rotation (upper table) and the unidimensional model (lower table).

	F1	F2	F3
CCAS1	−0.66	−0.39	-
CCAS2	−0.62	−0.65	-
CCAS3	−0.49	−0.47	0.22
CCAS4	−0.53	−0.38	0.27
CCAS5	−0.55	-	0.45
CCAS6	−0.69	-	0.51
CCAS7	−0.39	-	0.27
CCAS8	−0.50	-	0.50
CCAS9	−0.82	-	-
CCAS10	−0.61	-	-
CCAS11	−0.85	-	-
CCAS12	−0.83	-	-
CCAS13	−0.64	-	-
	**F1**	**F2**	**F3**
Explained variance	41.4%	7.4%	7.3%
	**F1**		
CCAS1	0.73		
CCAS2	0.70		
CCAS3	0.58		
CCAS4	0.63		
CCAS5	0.64		
CCAS6	0.77		
CCAS7	0.45		
CCAS8	0.58		
CCAS9	0.79		
CCAS10	0.60		
CCAS11	0.81		
CCAS12	0.80		
CCAS13	0.65		
	**F1**		
Explained variance	46%		

**Table 5 ijerph-21-01158-t005:** Comparison of loadings in the unidimensional model and the bifactor model.

	Unidimensional Model	Bifactorial Model		
	F	G	CEI	FI
CCAS1	0.727	0.721	0.300	-
CCAS2	0.700	0.735	0.539	-
CCAS3	0.579	0.615	0.355	-
CCAS4	0.631	0.661	0.250	-
CCAS5	0.635	0.690	−0.150	-
CCAS6	0.766	0.836	−0.215	-
CCAS7	0.445	0.464	−0.179	-
CCAS8	0.581	0.650	−0.254	-
CCAS9	0.790	0.687	-	0.443
CCAS10	0.597	0.525	-	0.315
CCAS11	0.811	0.690	-	0.537
CCAS12	0.803	0.691	-	0.502
CCAS13	0.651	0.592	-	0.253

**Table 6 ijerph-21-01158-t006:** Comparison of variance explained between the unidimensional model and the bifactorial model.

	Unidimensional Model	Bifactorial Model		
	F	G	CEI	FI
Explained variance	46%	44.2%	5.7%	6.9%

**Table 7 ijerph-21-01158-t007:** Discrimination and difficulty parameters for CCAS items *n* = 905.

	*a*	*b* _1_	*b* _2_	*b* _3_	*b* _4_
CCAS1	1.8	−1.09	−0.05	1.19	2.63
CCAS2	1.67	−0.84	0.25	1.47	2.74
CCAS3	1.21	0.64	1.85	3.48	5.59
CCAS4	1.38	0.58	1.42	2.88	4.51
CCAS5	1.4	−0.86	0.08	1.03	2.71
CCAS6	2.03	−0.29	0.64	1.51	2.84
CCAS7	0.84	0.86	1.94	3.3	5.7
CCAS8	1.22	−0.15	0.95	2.06	4.09
CCAS9	2.19	−0.52	0.37	1.49	2.56
CCAS10	1.27	−1.57	−0.53	0.76	2.6
CCAS11	2.36	−0.22	0.66	1.53	2.51
CCAS12	2.3	−0.15	0.63	1.45	2.34
CCAS13	1.46	−0.41	0.5	1.44	2.78

## Data Availability

Data are contained within the article.

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
