# Peer review of "A Follow up on the Continuum Theory of Eco-Anxiety: Analysis of the Climate Change Anxiety Scale Using Item Response Theory among French Speaking Population"

_ijerph, 2024, doi:10.3390/ijerph21091158_

Round 1

Reviewer 1 Report

Comments and Suggestions for Authors

Comments and Suggestions for Authors

I appreciate the opportunity to review this interesting article on the “A follow up on the continuum theory of eco-anxiety: Analysis of the Climate Change Anxiety Scale using Item Response Theory.” I enjoyed reading the article. I recommend the authors to improve the strengths of their article including:

Title: A good title should include the location of the study. I suggest the authors to add the study area in the title.

Abstract:

- (Lines 9-10) Please rephrase this sentence “we use item response theory (IRT) to understand better what existing questionnaires measure” to reflect the objective of this study. You can use other sentences such as “This study aims…..” or “The aims of this study…..” or any relevant sentence.

- (Line 11) You have mentioned the methodology on “open-access data from the short French version of the Climate Change Anxiety Questionnaire”. Please briefly describe the methodology on Climate Change Anxiety Scale and data analysis using MIRT package, based on the Graded Response Model.

- (Line 12) Please remove the citation.

1. Introduction

- (Line 24) “However, there is a heterogeneity in the way it is defined [1, 2, 3].” => Please rephrase this sentence and replace the word “heterogeneity” with other appropriate words.

- The word ‘climate’ is first introduced in line 31 in the Introduction. I suggest the authors to include an explanation of the mechanism of how climate change will impact the anxiety of people, so that there is a connection between the previous sentence in 30 and the current sentence in line 31.

- (Lines 35-38) “Thus, not all measurement tools incorporate in their development the idea of a continuous conception of eco-anxiety.” Briefly explain the advantages and disadvantages of the measurement tools that you are referring to. Thus, you can express the contribution of your study based on the developed measurement tools.

- (Lines 41-43) Please add the citation for this sentence => “Several authors have attempted to quantify this psychological discomfort linked to climate change and the environmental crisis in other ways, but no conceptualization takes into account the full spectrum of this discomfort.”

- (Lines 112-113) Please move this sentence to the last paragraph in the Introduction “The aim of this article is to provide some keys to clarifying these fuzzy areas relating to the conceptualization and measurement of eco-anxiety through the use of IRT”.

- Please re-organize your sentences in the Introduction. A few sentences in the Introduction should move to the Methodology. For example, sentences in lines 149-154.

2. Materials and Methods

- (Line 171-173) Why do you choose 905 and 873 respondents? Is it based on any formula? Please give a justification on why you use this number of respondents. Please describe the location or study area of the respondents.

- (Line 175) Please briefly describe the configurations used for the MIRT package.

- (Lines 198 – 208) The authors have mentioned the acceptable range for a few indicators that reflect the accuracy of your results such as RMSEA, SRMR, CFI, and TLI. However, the range for AIC is missing. Please include acceptable range for AIC to compare the fitness of the models

3. Results

- (Line 233) The authors should start describing the overview of the Results in a few sentences, which at least in one paragraph. The authors should not directly jump into a sentence that describes Table 1.

- (Lines 236-237) Table 1 => Please describe in detail the comparison of the findings for each model, which comprises of 1 factor, 2 factors, 3 factors, and bifactor.

- (Lines 253-256) Table 2 => Briefly mention in detail the differences of the 1 to 2 and 3 dimensional models and the two-factor model, by referring to AIC, χ², df and p.

- Tables 3 (lines 272-277) and table 4 (280-284) => Please describe the differences of the explained variance for FI, SYM and RUM.

- Please describe on how you validate the models developed in your study.

4. Discussion

- (Lines 331-332) Please include the goodness of fit of the unidimensional model that has been found in the literature, and compare with the findings of your study.

- Please include the comparison of the models developed in your study with the models developed by other studies. Explain the novelty of your study in terms of the contribution of new knowledge, methodology, and theory.

- Explain the benefits of this study to the current policy and relevant agencies.

5, Conclusion

- (Line 380 and 386) Please remove the citation.

- Conclusion should express the summarization and synthesis of the main findings of your study, not the review of the previous literature. Please remove sentences in lines 380-382 and rephrase them with appropriate sentences which show the summarization and synthesis of your main findings.

Comments on the Quality of English Language

Minor editing of English language required.

Author Response

Dear reviewer, thank you for your positive feedback and for your on-point comments. I hope you find my replies to them satisfactory.

Title :

Comment 1 : A good title should include the location of the study. I suggest the authors to add the study area in the title.

Response 1 : thank you for the suggestion. Therefore th we added “ among French speaking Europeans” to the title (line 4)

Abstract:

Comment 2 : (Lines 9-10) Please rephrase this sentence “we use item response theory (IRT) to understand better what existing questionnaires measure” to reflect the objective of this study. You can use other sentences such as “This study aims…..” or “The aims of this study…..” or any relevant sentence.

Response 2 : Thank you for pointing this out. Sentence rephrased to “. This study aims to utilize item response theory (IRT) to gain a deeper understanding of the con-structs measured by existing questionnaires.” (line 9 -10)

 Comment 3: (Line 11) You have mentioned the methodology on “open-access data from the short French version of the Climate Change Anxiety Questionnaire”. Please briefly describe the methodology on Climate Change Anxiety Scale and data analysis using MIRT package, based on the Graded Response Model.

Response 3 :  we added to the abstract a sentence to mention the statistical analysis (lines 13 to 15)

Comment 4 :  (Line 12) Please remove the citation.

Response 4 : citation deleted.

  1. Introduction

Comment 5 :  (Line 24) “However, there is a heterogeneity in the way it is defined [1, 2, 3].” => Please rephrase this sentence and replace the word “heterogeneity” with other appropriate words.

Response 5 : sentence rephrased to : “However, its definition may vary depending of the context of the study and the concepts used to quantify it” (lines 27,28)

Comment 6 : The word ‘climate’ is first introduced in line 31 in the Introduction. I suggest the authors to include an explanation of the mechanism of how climate change will impact the anxiety of people, so that there is a connection between the previous sentence in 30 and the current sentence in line 31.

Response 6 : thank you for pointing this out. Due to a mistake, there was supposed to be a note explaining climate change anxiety. This was added to the main text now. From line 32 to 45 we discussed the difference between eco-anxiety and climate anxiety.

Comment 7 : (Lines 35-38) “Thus, not all measurement tools incorporate in their development the idea of a continuous conception of eco-anxiety.” Briefly explain the advantages and disadvantages of the measurement tools that you are referring to. Thus, you can express the contribution of your study based on the developed measurement tools.

Response 7 : later in the text we explain that measure tools suffer from a floor effect with makes them bad at differentiating between respondents with mild forms of eco-anxiety but it is a problem that is inherent to the fact that these questionnaires are trying to capture clinical forms of eco-anxiety. Lines 72 to 78 are also added to sum up what we think are the advantages and disadvantages of such questionnaires.

Comment 8 : (Lines 41-43) Please add the citation for this sentence => “Several authors have attempted to quantify this psychological discomfort linked to climate change and the environmental crisis in other ways, but no conceptualization takes into account the full spectrum of this discomfort.”

Response 8 : citations added.

Comment 9 : (Lines 112-113) Please move this sentence to the last paragraph in the Introduction “The aim of this article is to provide some keys to clarifying these fuzzy areas relating to the conceptualization and measurement of eco-anxiety through the use of IRT”.

Response 9 : sentence moved

Comment 10 : Please re-organize your sentences in the Introduction. A few sentences in the Introduction should move to the Methodology. For example, sentences in lines 149-154.

Response 10 : on going

  1. Materials and Methods

Comment 11 :  (Line 171-173) Why do you choose 905 and 873 respondents? Is it based on any formula? Please give a justification on why you use this number of respondents. Please describe the location or study area of the respondents.

Response 11 : as you may have seen the data used in this article come form 2 other studies. On the first study the researchers collected 905 responses on the CCAS questionnaire. On the second they collected 873. We took the entire data frames to use in our study. No formula was used to select these numbers. Lines from 196 to 199 were added to describe the studies area.

Comment 12 : (Line 175) Please briefly describe the configurations used for the MIRT package.

Response 12 : configuration described in lines 204 to 208 R studio script can be provided on demand.

Comment 13 : (Lines 198 – 208) The authors have mentioned the acceptable range for a few indicators that reflect the accuracy of your results such as RMSEA, SRMR, CFI, and TLI. However, the range for AIC is missing. Please include acceptable range for AIC to compare the fitness of the models

Response 13 : There is no cut of criteria for AIC parameter. The value of this parameter should be lower for statistically better models. Sentence changed to “The lower the value of AIC compared to previous models the better the model is”( line 237-238)

  1. Results

Comment 14 : (Line 233) The authors should start describing the overview of the Results in a few sentences, which at least in one paragraph. The authors should not directly jump into a sentence that describes Table 1.

Response 14 : table 1 was moved down after the text where it was in our opinion best suited. table 1 contains all the results for the model fit statistics for all the models tested in this study. Reading the results based on the criterion provided in “materiel and method” provides description of the model fit of each model which is completed by table 2 where we compare the significance of improvement for each model. 

Comment 15 : (Lines 236-237) Table 1 => Please describe in detail the comparison of the findings for each model, which comprises of 1 factor, 2 factors, 3 factors, and bifactor.

Response 15 : I am sorry to disagree with you on this point. based on pour response 14, I estimate that describing in detail the comparison of the findings for each model will be redundant with table 1 and table 2. In the text I think that we discussed all the statistical results that are relevant to our argument. If you disagree with our response, we humbly ask you to provide more insight on how we can improve this part. Lines 262 to 271 were added to give better insight on the results.

Comment 16 : (Lines 253-256) Table 2 => Briefly mention in detail the differences of the 1 to 2 and 3 dimensional models and the two-factor model, by referring to AIC, χ², df and p.

Response 16 : a paragraph (line 264-271) was added to try to answer comment 15 and 16

Comment 17 : Tables 3 (lines 272-277) and table 4 (280-284) => Please describe the differences of the explained variance for FI, SYM and RUM.

Response 17 : Dear reviewer, taken the arguments we provide regarding the impact of the choice of rotation on the interpretation the factors. We think that the description of the explained variance for each factor seems irrelevant here because we give arguments to refuted. Explanation for the problematic use of rotation in factor analysis and interpretation are provided in the discussion (lines 378 to 395). We judge that this paragraph is better suited for discussion please notify us if you think otherwise.

Comment 18 : Please describe on how you validate the models developed in your study.

Response 18 : Dear reviewer, the results section, is supposed to be a step by step evaluation of alternative models for the CCAS un an exposition of the limits of each one tested to lead to the retention of the unidimensional model. We did our best to reflect this idea on the text. We also added lines 341 to 346 to summarize the findings and justify the unidimensional selection. Please feel free to notify us if this section needs to be improved further.

  1. Discussion

Comment 18 : (Lines 331-332) Please include the goodness of fit of the unidimensional model that has been found in the literature, and compare with the findings of your study.

Response 18 : lines 379 to 382 are added to the text

Comment 19 : Please include the comparison of the models developed in your study with the models developed by other studies. Explain the novelty of your study in terms of the contribution of new knowledge, methodology, and theory.

Parts that are added to emphasize the novelty of this study are highlighted in red in the discussion 

Comment 20 :  Explain the benefits of this study to the current policy and relevant agencies.

Comment 20 : lines 440 to 443 are added to answer this suggestion.

5, Conclusion

Comment 21 : (Line 380 and 386) Please remove the citation.

Response 21 : Limitations and discussion were added to the discussion. Thus we kept the citations

Comment 22 : Conclusion should express the summarization and synthesis of the main findings of your study, not the review of the previous literature. Please remove sentences in lines 380-382 and rephrase them with appropriate sentences which show the summarization and synthesis of your main findings.

Response 22 : a new conclusion had been written (lines 460 – 481)

Reviewer 2 Report

Comments and Suggestions for Authors

Dear Authors,

I read your research with high interest. I congratulate you on the research and the way you presented the results. 

I have only a few suggestions regarding the conclusions. Line 380, you say this idea, but it is vague. I think you should state that idea in the text. 

Also, I think it will be interesting to add a subsection to emphasize the theoretical and practical implications of your research at various levels: individuals, countries, global, companies, health sector, environment and so on. 

Wish you great success with your research. 

Author Response

Dear reviewer thank you for the interest you gave to my article and for your positive feedback.Here is a reply to your comments.For your comment on line 380 (new line 440) we changed the word idea with "the continuum hypothesis of climate anxiety". We also added to the discussion a section to discuss more explicitly about the implications of our study for different concerned parties (lines 451 to 458). We hope it is satisfactory for you. We are more than happy to receive further feedbacks from you. 

Round 2

Reviewer 1 Report

Comments and Suggestions for Authors

I read the manuscript again. I appreciate the corrections made by the authors. Now the article has more scientific features and seems more interesting. I recommend the authors to improve the flow of their academic writing including:

In my previous review, I mentioned re-organizing your sentences in the Introduction. A few sentences in the Introduction should move to the Methodology. The author answered my comment with "on going". Please improve this part and I am waiting for your revision.

Author Response

Dear Reviewer,

Thank you once again for your interest in our work and the attention you have given it.

We have made efforts to enhance the overall quality of the manuscript and hope that the writing now meets your standards. We are grateful for your guidance in refining our manuscript.

As part of our revisions, we have relocated certain paragraphs from the introduction to the methodology section. Specifically, we removed the lines from 133 to 136 and incorporated them into the first paragraph of the methodology section (now lines 169 to 180). Additionally, we moved the paragraph discussing the models tested (originally lines 171 to 180) and Figure 1 to the fourth paragraph of the "Materials and Methods" section (now lines 195 to 205).